# One Scaffold, Two Conformations: The Ring-Flip of the Messenger InsP_8_ Occurs under Cytosolic Conditions

**DOI:** 10.3390/biom13040645

**Published:** 2023-04-04

**Authors:** Leonie Kurz, Peter Schmieder, Nicolás Veiga, Dorothea Fiedler

**Affiliations:** 1Leibniz-Forschungsinstitut für Molekulare Pharmakologie, Robert-Rössle-Straße 10, 13125 Berlin, Germany; 2Institut für Chemie, Humboldt-Universität zu Berlin, Brook-Taylor-Straße 2, 12489 Berlin, Germany; 3Química Inorgánica, Departamento Estrella Campos, Facultad de Química, Universidad de la República (UdelaR), Av. Gral. Flores 2124, Montevideo 11800, Uruguay

**Keywords:** InsP_8_, inositol pyrophosphates, conformation, NMR, ring-flip, molecular switch, pH sensing, metal coordination

## Abstract

Inositol poly- and pyrophosphates (InsPs and PP-InsPs) are central eukaryotic messengers. These very highly phosphorylated molecules can exist in two distinct conformations, a canonical one with five phosphoryl groups in equatorial positions, and a “flipped” conformation with five axial substituents. Using ^13^C-labeled InsPs/PP-InsPs, the behavior of these molecules was investigated by 2D-NMR under solution conditions reminiscent of a cytosolic environment. Remarkably, the most highly phosphorylated messenger 1,5(PP)_2_-InsP_4_ (also termed InsP_8_) readily adopts both conformations at physiological conditions. Environmental factors—such as pH, metal cation composition, and temperature—strongly influence the conformational equilibrium. Thermodynamic data revealed that the transition of InsP_8_ from the equatorial to the axial conformation is, in fact, an exothermic process. The speciation of InsPs and PP-InsPs also affects their interaction with protein binding partners; addition of Mg^2+^ decreased the binding constant K_d_ of InsP_8_ to an SPX protein domain. The results illustrate that PP-InsP speciation reacts very sensitively to solution conditions, suggesting it might act as an environment-responsive molecular switch.

## 1. Introduction

Myo-inositol polyphosphates (InsPs) are a family of messenger molecules that control a wide array of biological processes in eukaryotic cells (Figure 1a). In these molecules, the phosphoryl groups are arranged in different numbers and patterns around the inositol scaffold, creating great structural variety that ranges from species with only one phosphoryl group to the inositol pyrophosphates (PP-InsPs), which carry seven or eight phosphates [1,2,3]. The PP-InsPs contain one or two high-energy diphosphate groups, in addition to monophosphates, thereby accommodating an extraordinary negative charge density.

The most highly phosphorylated PP-InsP in mammals is 1,5(PP)_2_-InsP_4_, (from here on abbreviated as InsP_8_), which is generated via the phosphorylation of 5PP-InsP_5_ by PPIP5Ks (diphosphoinositol-pentakisphosphate kinases), or phosphorylation of 1PP-InsP_5_ by IP6Ks (inositol hexakisphosphate kinases). In recent years, InsP_8_ has emerged as a regulator of inorganic phosphate homeostasis in various organisms. The enzymatic activity of PPIP5Ks is directly regulated by phosphate concentrations, and cellular levels of InsP_8_ correlate with phosphate availability [4,5]. In fission yeast (*S. pombe*), InsP_8_ activates the vacuolar VTC complex by putatively binding to an SPX domain, driving polyphosphate synthesis to store excess phosphate [6]. In plants, it was demonstrated that the activity of the InsP_8_-producing enzymes VIH1/2 negatively regulates phosphate starvation responses [7]. Mechanistically, InsP_8_ binds to a standalone SPX domain (SPX1) in *A. thaliana*, which enables dimerization with the transcription factor PHR1 and suppresses downstream targets [8]. In cultured human cell lines, InsP_8_ was found to activate phosphate efflux by binding to the SPX domain of XPR1, removing excess phosphate and preventing tissue calcifications [9,10,11]. Moreover, at an organismal level, mutations in PPIP5Ks are associated with hearing loss [12,13].

The function of 5PP-InsP_5_ has been investigated more closely, especially its role in cellular energy signaling, where the production of 5PP-InsP_5_ critically depends on ATP levels and regulates glycolytic flux accordingly [14,15]. Notably, 5PP-InsP_5_ stimulates exocytosis of insulin-containing granules from pancreatic β-cells, ostensibly by competing with the plasma membrane lipid phosphatidylinositol-4,5-bisphosphate (PIP_2_) for binding of synaptotagmin 7 [16,17]. 5PP-InsP_5_ also inhibits the activity of Akt kinase, a central metabolic regulator, by binding to the PH domain of Akt, releasing it from the plasma membrane, preventing its activation [18,19,20]. As a result, IP6K1 knockout mice display abnormally high Akt activity and are resistant to weight-gain when fed a high-fat diet [18]. IP6K1 knockout in mice also increases the activity of AMPK, inhibiting fat accumulation in favor of thermogenesis, and further supporting the lean phenotype [21].

In many cases it is difficult to attribute an observed phenotype to an individual messenger, because genetic deletion of IP6Ks inevitably reduces the levels of both 5PP-InsP_5_ and InsP_8_ [22]. Even when biochemical insight into the mechanisms of action is available, it sometimes cannot explain why PP-InsPs possess distinct biological functions, in spite of their close structural relatedness and similarly high charge density. For example, phosphate efflux via XPR1 is activated almost exclusively by InsP_8_, even though 1PP-InsP_5_ and 5PP-InsP_5_ bind with similar dissociation constants (K_d_) to the SPX domain [9]. Binding affinities are also similar for InsP_8_ and 5PP-InsP_5_ towards the SPX1 protein in rice (*O. sativa*), but genetic experiments have indicated that only InsP_8_ is the physiologically relevant ligand [8]. Similarly, in in vitro experiments, InsP_8_ is twenty-times more effective than 5PP-InsP_5_ at activating polyphosphate synthesis by the VTC complex from budding yeast *S. cerevisiae*, despite a charge difference of only one. It was also more effective than other InsP_8_ isomers [23], suggesting that affinity is determined by the exact shape of the molecule, rather than sheer charge density. In another example from fission yeast *S. pombe*, polyphosphate synthesis by the VTC complex clearly depended on InsP_8_, synthesized by the PPIP5K orthologue Asp1 [6]. Possible explanations for how substrate specificity of PP-InsP-binding proteins may be regulated include ternary interactions with more than one binding partner, local production of individual PP-InsPs, and localized differences in solution conditions, which change protonation and metal complexation.

A property unique to highly phosphorylated InsPs is their ability to adopt an alternate conformation at elevated pH, where the substituent at the 2-position is equatorial, and all others are axial, which separates the phosphoryl groups further in space and reduces charge repulsion between them (Figure 1b,c) [24,25,26,27]. The PP-InsPs and their non-hydrolyzable methylene bisphosphonate analogs (PCP-InsPs, Figure 1b) also display this behavior, and appear to have a higher propensity to adopt the axial conformation relative to the InsPs, especially in the presence of magnesium cations [28].

PP-InsPs can potentially form a wide range of different species, depending on their conformation, protonation state and complexation of metal cations. We therefore wanted to obtain a more detailed understanding of PP-InsP speciation, specifically under conditions approximating a cytosolic setting. Due to the structural information it provides, NMR spectroscopy is the method of choice for characterizing these equilibria, but the lack of sensitivity has limited past investigations to experiments using non-physiological concentrations of InsPs and PP-InsPs (>100 µM) [28].

Here, we have conducted a systematic comparison of InsP_6_, 5PP-InsP_5_ and InsP_8_ using NMR spectroscopy of ^13^C-labeled compounds, to better understand the intricacies of their speciation. Intriguingly, we found that InsP_8_ is able to adopt the axial conformation under conditions very reminiscent of a cytosolic environment. We then used ^31^P-NMR to identify likely protonation states and K^+^- and Mg^2+^-complexes of InsP_8_. Finally, ITC experiments revealed that addition of Mg^2+^ ions influenced the binding parameters of the interaction between InsP_8_ and an SPX protein domain. Our results imply that conditions for biochemical and biophysical characterization of PP-InsP protein interactions should always be chosen with great care, and highlight InsP_8_ as a potential pH-, metal ion-, and temperature-dependent intracellular molecular switch.

## 2. Materials and Methods

### 2.1. Synthesis and BIRD-HMQC NMR Analysis of InsPs

^13^C-labeled InsPs and PP-InsPs were synthesized chemo-enzymatically, as previously published, with slight adjustments [30,33]. Enzymatic synthesis of InsP_8_ was carried out at pH 6.0 instead of 6.4, and in the presence of additional 150 mM (NH_4_)_2_SO_4_.

Samples for HMQC-NMR studies contained 50 µM ^13^C_6_-labeled InsPs (InsP_6_/5PP-InsP_5_/InsP_8_), 2 mM bis-tris-propane, 130 mM KCl, 10 mM NaCl and 0/50/250 µM MgCl_2_ in D_2_O.

BIRD-HMQC NMR spectra were recorded at 277 K and 600 MHz (^1^H frequency) on a Bruker AV-III NMR spectrometer (Bruker Biospin, Rheinstetten, Germany) using cryogenically cooled 5 mm QCI-triple resonance probe equipped with one-axis self-shielded gradients. The spectrometer was operated using topspin 3.5 pl6 software. Instrument temperature was calibrated against _4_-methanol according to Findeisen et al. [34]. Acquisition parameters were SW(^13^C): 60–90 ppm, NS: 128, TD(^13^C): 64.

### 2.2. Assignment of HMQC-NMR Spectra of Axial Conformations

BIRD-HMQC spectra were recorded at 277 K as described above, with samples containing 200 µM InsPs, 130 mM KCl, 10 mM NaCl, 2mM BTP in D_2_O. pH was set to pH* 9.5 for InsP_6_, pH* 8.8 for 5PP-InsP_5_ and pH* 8.5 for InsP_8_ (pH = 0.929 × pH* + 0.42 [35]). NOESY-HMQC spectra of the same samples were subsequently recorded using mixing times of 80 ms for InsP_6_ and 5PP-InsP_5_ and 200 ms for InsP_8_. Conformational exchange on the time-scale of the experiment created cross-peaks between corresponding peaks in different conformations, which allowed us to assign each peak of the equatorial conformer to its axial counterpart.

### 2.3. Van’t Hoff Thermodynamic Analysis

BIRD-HMQC NMR spectra were recorded as described above, at 274–283 K in 1 K steps, two replicate samples per condition. Instrument temperature was calibrated against d_4_-methanol according to Findeisen et al. [34].

Samples contained 50 µM InsP_8_, 5 mM HEPES pH* 7.5 (pH set on ice), 130 mM KCl, 10 mM NaCl, and either 100 µM or 250 µM MgCl_2_. Acquisition parameters were SW(^13^C): 62–82 ppm, NS: 2048, TD(^13^C): 16.

The best-isolated peaks (the 2-peak of ax. and 1-peak of eq. conformer) were integrated, subtracting 1PP-InsP_5_ impurities. The ax./eq. ratio (i.e., equilibrium constant K for the eq. to ax. transition) was plotted against 1/T, where T is temperature in Kelvin.

ΔH^0^ and ΔS^0^ were calculated from the regression line. Two replicates were combined for the linear regression.

### 2.4. NMR Titrations

In order to create a system free of coordinating counterions, a batch of InsP_8_ was synthesized as described above, by enzymatic phosphorylation of 5PP-InsP_5_ and precipitation with Mg^2+^_,_ followed by Mg^2+^ chelation on Amberlite^®^ cation exchange resin. Unlike in the standard procedure, the resin was loaded with NMe_4_Cl instead of NH_4_CO_3_, resulting in a batch of InsP_8_ with non-coordinating NMe_4_^+^ counterions. 1 mM EDTA was added to all ^31^P-NMR titrations to chelate remaining traces of Mg^2+^.

NMR-samples contained 1 mM InsP_8_, 1 mM EDTA, pH 3.0–12.5 (in H_2_O, steps of 0.5 pH units) and a) for the non-coordinating condition: 150 mM NMe_4_Cl or b) 150 mM KCl or c) 150 mM KCl and 1 mM MgCl_2_. Sealed glass capillaries containing 50 mM phosphonoacetic acid in D_2_O were added into the sample tubes for locking and chemical shift calibration.

^31^P-NMR spectra were recorded at 295 K on a Bruker spectrometer (see above) operating at 600 MHz for protons and 244 MHz for phosphorous nuclei. SW: −40 ppm to −20 ppm, NS: 1024. Spectra at pH 3.0 were assigned by various 2D-NMR methods (Appendix A) and chemical shift changes were tracked across the pH range. 

The data were analyzed using the HypNMR 2006 software (Hyperquad Limited, Lincoln, UK) [36]. Different possible stoichiometries were tested, and the final chemical models were selected on the basis of the σ parameter (scaled sum of square differences between predicted and experimental chemical shift values), the model confidence level estimator (chi square), and the internal consistency of data reflected in standard deviations of the formation constants [37]. Species distribution diagrams were produced using the HySS program (Hyperquad Limited, Lincoln, UK) [38].

### 2.5. Isothermal Titration Calorimetry

The VTC2 SPX domain (residues 1–182) was expressed with a C-terminal His_6_-tag and purified by Ni-affinity and size-exclusion chromatography, as previously published [39].

Protein stocks were diluted to 300 µL with final buffer conditions: 25 mM HEPES pH 7.4, 150 mM KCl, 40 mM NaCl, 0.5 mM TCEP (ITC buffer). The exact protein concentration for each replicate was determined separately by Bradford assay. InsP_8_ was diluted to 500 µM in ITC buffer. For binding experiments in the presence of magnesium ions, both dilution buffers were supplemented with MgCl_2_ to give a final concentration of 1 mM after dilution.

ITC experiments were carried out at 25 °C in a MicroCal PEAQ-ITC calorimeter (Malvern Panalytical GmbH, Kassel, Germany), with ca. 50 µM protein in the cell and 500 µM ligand in the syringe. InsP_8_ was titrated into the solution in nineteen 2 µL-steps. Spacing between injections was 150 s.

The corresponding instrument software (MicroCal PEAQ-ITC Analysis) was used for baseline correction, peak integration, data fitting and determination of binding parameters.

### 2.6. DFT Calculations

The input geometries were built employing the protonation and complexation patterns determined by ^31^P NMR in this report. Three water molecules were included in the first coordination sphere of the Mg^2+^ cation. The initial geometries were pre-optimized by means of a molecular mechanic method (MM+), in order to explore the potential energy surface. Then, a Density Functional Theory (DFT) optimization protocol was performed in Gaussian 09 (Gaussian Inc., Wallingford CT, USA) [40], using the B3LYP functional, with an ultrafine integration grid and the 3–21 + G* split valence basis set. The potassium ions were treated using the effective core potential LANL2DZ relativistic procedure [41]. The solvent was modelled through the Truhlar and coworkers’ SMD solvation model [42]. All final structures were minima in the potential energy surface, being the nature of the stationary points verified through vibrational analysis.

## 3. Results

### 3.1. ^1^H,^13^C-HMQC NMR Spectra of PP-InsPs Can Show Two Conformations Simultaneously

The recent development of enzymatic syntheses for inositol pyrophosphates provides access to ^13^C-labeled PP-InsPs at the scale of hundreds of micromoles [30,33]. The uniformly labeled PP-InsPs can readily be detected at concentrations below 50 µM in ^1^H,^13^C HMQC experiments, which motivated us to systematically investigate InsP_6_/PP-InsP speciation and conformation under different conditions, near physiological concentrations.

A ^1^H,^13^C HMQC NMR spectrum of InsP_6_ (50 µM, no coordinating counter ions present) at pH* 6.5 (pH* = apparent pH in D_2_O [35]; pH* 6.50 = pH 6.46) displayed four peaks due to the molecule’s plane of symmetry, and was assigned to correspond to the equatorial conformation of InsP_6_ (five phosphoryl groups in equatorial position, position 2 in axial position). An equivalent sample at pH* 12.0 displayed a different set of four peaks, shifted upfield in the carbon dimension, which corresponds to the axial conformation of InsP_6_ (five phosphoryl groups axial, position 2 is equatorial) [43,44].

Unfortunately, when recording spectra at 37 °C, we noticed NMR-peak broadening, making detection of InsP_6_ conformers impossible in a certain pH-range. This intermediate exchange effect had been observed before and impedes direct observation of InsPs/PP-InsPs by proton-based NMR methods near the conformational transition [26,28]. We therefore attempted to slow down exchange rates by cooling the sample and recording spectra at 4 °C. An HMQC-spectrum at 4 °C and pH* 9.0 (=pH 8.8) displayed signals of both the axial and the equatorial conformer simultaneously (Figure 2a).

Concurrent observation of both conformations was also possible for the inositol pyrophosphates 5PP-InsP_5_ and InsP_8_ at 4 °C (Figure 2b,c). All HMQC–NMR peaks of IP_6_ and the PP-InsPs in the axial conformation were assigned using a NOESY–HMQC–NMR method, which creates cross-peaks between corresponding peaks of the two conformations (Appendix A). These assignments allowed us to use integration to quantify the relative proportions of the two conformers.

### 3.2. Conformational Equilibria of InsP_6_ and PP-InsPs Are Sensitive to pH and Ionic Composition

With the ability to detect and quantify both conformations over a wide pH range, at low InsP/PP-InsP concentrations, we sought to systematically compare the behavior of InsP_6_, 5PP-InsP_5_ and InsP_8,_ and characterize the influence of pH and biologically relevant metal ions on their conformation. We decided to maintain physiological concentrations of K^+^ and Na^+^ ions, while varying pH and concentration of the divalent cations Mg^2+^ and Ca^2+^.

Since inositol polyphosphates are prone to precipitation in the presence of divalent metal ions, especially at basic pH, we wanted to ensure that this phenomenon did not perturb our analysis. In the presence of 5 equiv. Mg^2+^, all three molecules showed notable precipitation within 24 h at pH* 9.0, but not at pH* 8.0 and lower (Appendix A).

Without divalent cations present, InsP_6_ remained in its equatorial conformation over most of the observed pH range (6.5–9.5). Only when pH* was increased to 9.0 did traces of the axial conformer begin to appear, and even at pH* 9.5, less than half of InsP_6_ was in the axial conformation (Figure 3a). The proportion of the axial conformer was increased by addition of Mg^2+^ cations. Addition of one equivalent was enough to shift the equilibrium to about 30% axial conformer at pH 9.0* and 90% axial conformer at pH* 9.5. Addition of five equivalents of Mg^2+^ facilitated the transition to the axial conformer even more (Figure 3a). Similar results were obtained when adding Ca^2+^ (Appendix A). These observations are consistent with previous reports that an elevation of pH and addition of metal cations can increase the proportion of the axial conformer [26,28,45].

For 5PP-InsP_5_, without divalent cations present, about 10% axial conformer were already detected at pH* 8.5. At pH* 9.5, 5PP-InsP_5_ was only detected in the axial conformation (Figure 3b). Like InsP_6_, the proportion of 5PP-InsP_5_ in the axial conformation was increased upon addition of Mg^2+^ ions. In the presence of five equivalents Mg^2+^, 5PP-InsP_5_ adopted exclusively the axial conformation at pH* 8.5 (=pH 8.3) and higher (Figure 3b). The effect of Ca^2+^ was slightly less pronounced. About 60% of 5PP-InsP_5_ were in an axial conformation at pH* 8.5 in the presence of 5 equiv. Ca^2+^ (Appendix A).

The trend that transition to the axial conformation was facilitated by additional phosphoryl groups continued for InsP_8_. Even without divalent cations present, about half of the molecules adopted the axial conformation at pH* 8.5 (Figure 3c). The addition of Mg^2+^ ions again further shifted the conformational equilibrium towards the axial conformation. With one equivalent of Mg^2+^, about 10% of InsP_8_ were in the axial conformation at pH* 7.5 (=pH 7.4). Upon addition of five equivalents of Mg^2+^, InsP_8_ was almost equally distributed between its axial and equatorial forms at pH* 7.5. This last result is particularly interesting, because it suggests that a substantial portion of InsP_8_ can adopt the axial conformation under cytosolic conditions. Again, the effect of Ca^2+^ was similar but slightly weaker. About 15% of InsP_8_ adopted the axial conformation at pH* 7.5 with 5 equiv. Ca^2+^.

In sum, the characterization of InsP_6_ and PP-InsPs using ^1^H,^13^C-HMQC NMR was consistent with previous observations: The proportion of axial conformer rises with increasing pH and elevated concentrations of divalent cations. Moreover, the more densely phosphorylated the InsP/PP-InsP, the lower the pH at which the molecule begins to transition to the axial conformation. Each successive phosphorylation decreases the pH value for the transition by roughly 0.5 pH units. Intriguingly, InsP_8_ seems to undergo the conformational change at physiological pH and ionic composition, which prompted us to investigate this messenger molecule in more detail.

### 3.3. InsP_8_ Is Present in Both Conformations under Near-Physiological Conditions

To gain a deeper understanding of the forces governing the InsP_8_ conformational equilibrium under near-physiological conditions, we wanted to evaluate what proportion of axial conformer might be present at 37 °C, by determining the thermodynamic parameters of the conformational equilibrium (Figure 4a). To do so, we recorded HMQC-NMR spectra (50 µM InsP_8_, pH* 7.5, 130 mM KCl, 10 mM NaCl, 100 µM/250 µM MgCl_2_) over a temperature range of 10 K and measured the amounts of the two conformers via integration. It was not possible to use a wider temperature range, due to the peak broadening described above, and the necessity to detect both conformations. The values for ΔH^0^ and ΔS^0^ were extracted from slope and intercept of the van’t Hoff plots (Figure 4c,d). To assess the influence of Mg^2+^ ions, we measured two replicates with 2 equiv. and two replicates with 5 equiv. Mg^2+^, keeping all other parameters constant. To avoid precipitation, the Mg^2+^ concentration could not be increased any further, although the excess of Mg^2+^ ions is much higher in cells. Similarly, InsP_8_ could not be decreased to physiological concentrations of <1 µM without compromising the NMR detection.

In the presence of two equivalents of Mg^2+^, the equilibrium seems to be governed by a balance of enthalpic gains and entropic losses upon transition to the axial conformation (Figure 4b), with ΔH^0^ = −17.1 ± 2.1 kcal/mol and ΔS^0^ = −0.063 ± 0.008 kcal/(mol·K). Somewhat unexpectedly, the equatorial to axial transition of InsP_8_ is an exothermic process and the axial conformer is enthalpically more favorable. Using these thermodynamic values in the Gibbs-Helmholtz equation, we can estimate that at 37 °C, about 2% of InsP_8_ would adopt an axial conformation under these solution conditions.

With five equivalents Mg^2+^ present, the equatorial to axial transition was again an exothermic process, with ΔH^0^ = −7.2 ± 1.9 kcal/mol and ΔS^0^ = −0.025 ± 0.007 kcal/(mol·K). These values predict 29% axial conformer at 37 °C (310 K). Compared to the experiment with 2 equiv. MgCl_2_, both ΔH^0^ and ΔS^0^ were reduced in magnitude by more than half. The relative change of ΔS^0^ was larger, resulting in a net shift toward the axial conformation (Figure 4a,b).

In sum, our results suggest that substantial amounts of axial InsP_8_ could be formed under cytosolic conditions, and that the conformational equilibrium is largely governed by entropic effects.

### 3.4. InsP_8_ Forms Strong Complexes with Potassium and Magnesium Ions

Given the unexpected thermodynamic parameters for the conformational equilibrium, we next wanted to characterize the protonation sequence and metal complexation of InsP_8_ in solution. We utilized ^31^P NMR for detection, because ^31^P-NMR chemical shifts are very sensitive to both protonation and metal complexation, shifting upfield with each protonation step and downfield upon metal complexation [26,28,45]. The lower Larmor frequency of ^31^P compared to ^1^H allows detection of peaks which are broadened in ^1^H NMR due to intermediate exchange phenomena.

^31^P-NMR peaks were assigned to the eight phosphate groups (P1α, P1β, P2, P3, P4, P5α, P5β, P6) using a combination of 2D-NMR techniques (Appendix A). Titrations were performed at 1 mM InsP_8_ concentration in the pH-range 3.0–12.5 under three different conditions: (i) without coordinating cations, where ionic strength and pH were maintained with NMe_4_Cl/NMe_4_OH (Figure 5a), (ii) with 150 mM KCl (Figure 5b) and (iii) with 150 mM KCl and 1 mM MgCl_2_ (Figure 6a).

Under non-coordinating conditions, all signals shifted upfield with decreasing pH, as has been observed for InsP_6_ and 5PCP-InsP_5_ before (Figure 5a) [26,28]. Using HypNMR software, the experimental chemical shift values δ_P_ were fitted to generate a model of the protonation process and extract the first eight protonation constants of InsP_8_, starting from the fully deprotonated molecule (L^14−^, Table 1) [36]. Compared to InsP_6_ and 5PCP-InsP_5_, protonation constants were generally larger (indicating a greater proportion of protonated versus deprotonated state), presumably due to the higher negative charge of InsP_8_, which stabilizes higher protonation states. The optimized chemical model indicated that H_5_L^9−^ is the most abundant protonation state in the physiological pH range (Figure 5c and Appendix A). Based on the calculated protonation constants, theoretical δ_P_ values could be calculated, which were in excellent agreement with the experimental values. The theoretical δ_P_ values and protonation constants could then be used to determine Δδ_P_ values (changes of theoretical δ_P_ with each successive protonation step, Appendix A). Knowing that protonation causes an upfield shift, the Δδ_P_ values revealed the most likely protonation sites and sequence of protonation events (Appendix A). For example, during the first protonation step, the most negative Δδ_P_ values were observed for P3, P5β and P1β, suggesting that P3 shares its proton with P5β or P1β. Sudden chemical shift changes of all four monophosphate signals at around pH 11 indicated the conformational change (Figure 5a). Without coordinating cations present, the conformational change thus coincided with the second protonation step (for a full description of the analysis, see Appendix A).

Since InsP_8_ would never occur naturally without a background of coordinating ions, we repeated the ^31^P-NMR titration in the presence of 150 mM K^+^ ions (Figure 5b). Compared to the spectra recorded in solution devoid of coordination ions, all peaks were shifted downfield in the presence of K^+^, suggesting the formation of K^+^ complexes. Using HypNMR software, the protonation constants from above, and the experimental chemical shift values in the presence of K^+^, the formation constants and abundance of six different K^+^ complexes of InsP_8_ were calculated, along with the most likely sequence of protonation and complexation events (Figure 5d and Appendix A, Table 1). The [K_5_(HL)]^8−^ complex is by far the most abundant species down to about pH 10, reflected by the almost unchanging chemical shift values. The third protonation event (from [K_4_(H_2_L)]^8−^ to [K_4_(H_3_L)]^7−^) coincides with the conformational change, accompanied by a steep upfield shift of all monophosphate peaks around pH 8.5–9.0, consistent with our NMR data above (Figure 3).

Around physiological pH, the most abundant complex is predicted to be [K_2_(H_5_L)]^7−^, in which one K^+^ ion is coordinated to phosphates in position P6, P5α and P5β. The other K^+^ ion is coordinated to the pyrophosphate group at the 1-position, between P1α and P2 (Appendix A). Overall, K^+^ complexes of InsP_8_ were found to be substantially more stable than equivalent complexes with InsP_6_ or 5PCP-InsP_5_ (see formation constants in Table 1). For example, the formation constant of [K_2_(H_5_L)]^7−^, the main K^+^-complex of InsP_8_ at physiological pH, is more than tenfold higher than that of the corresponding 5PCP-InsP_5_ complex (for a full description of all detected complexes and their analysis, see Appendix A).

Finally, to more closely approximate physiological solution conditions and to include divalent cations, we recorded ^31^P-NMR titration curves of InsP_8_ in the presence of 150 mM KCl and 1 mM MgCl_2_ (Figure 6a). Combined with protonation and K^+^-complexation constants from the previous titrations, the formation constants and likely structures for seven K^+^-Mg^2+^ complexes were calculated (Table 1 and Appendix A). Due to the propensity of InsP_8_ to precipitate, it was not possible to add more than one equiv. MgCl_2_. The complex described above is therefore not entirely representative of biologically relevant species, but provides some insights nonetheless. Once again, Mg^2+^-complexes of InsP_8_ are far more stable than the equivalent ones formed by 5PCP-InsP_5_, as illustrated by much larger formation constants (Table 1).

The speciation diagram (Figure 6b) highlights the coexistence of several different complexes at any given pH value, especially towards lower pH. In the physiological pH range, the most abundant species is [MgK_3_(H_3_L)]^6−^, in which the Mg^2+^ ion is coordinated by the pyrophosphate group at 1-position, together with P2 (Figure 6c,d). As for the previous titrations, transition to the equatorial conformation was indicated by a steep upfield shift of monophosphate resonances, concomitant with the third protonation step at around pH 8.5 (likely in position 6, [K_3_Mg(H_2_L)]^7−^ to [K_3_Mg(H_3_L)]^6−^). At pH 8.0 and 8.5, the P2 peak was too broad to detect, suggesting higher rates of exchange between conformations in the presence of one equivalent Mg^2+^, which move the system into the intermediate exchange range even with ^31^P-NMR detection.

The results illustrate the extremely tight association between InsP_8_ and K^+^ or Mg^2+^ ions, and suggest an important role of the two pyrophosphate groups in Mg^2+^ binding. While in the axial conformation, InsP_8_ coordinates Mg^2+^ between its two pyrophosphate groups; once the conformation has changed to equatorial, the Mg^2+^ binding site is formed by the pyrophosphate group at position 1 and the phosphate group at position 2 (Figure 6c). These insights might inform future structural studies.

### 3.5. Complex Speciation of InsP_8_ Affects Protein Binding

Considering how sensitively InsP_8_ speciation reacts to solution conditions, such as pH and ionic composition, we wondered to what extent this speciation could influence the interaction of InsP_8_ with proteins. To test this, the binding of InsP_8_ to a known InsP-binding domain, the SPX domain (named after Syg1, Pho81, Xpr1 proteins) from yeast VTC2 [39,48] was investigated using isothermal titration calorimetry (ITC).

The first set of experiments was performed at pH 7.4 and approximately physiological salt composition, but without divalent ions. Based on our previous experiments, we expect InsP_8_ to adopt its equatorial conformation under these conditions. A fairly strong interaction between InsP_8_ and VTC2-SPX was observed, which could be fitted with a one-binding-site model (Figure 7a and Appendix A). The dissociation constant K_d_ was 373 nM (mean of three replicates), i.e., in the same range as previously published results for PP-InsPs binding to VTC2-SPX [39,48] or InsP_8_ interacting with the SPX domain of human XPR1 [9]. While both enthalpic and entropic parameters were in favor of binding, the entropic contribution was the dominant one (ΔH = −3.5 kcal/mol, −TΔS = −5.2 kcal/mol at 25 °C, mean of three replicates).

Next, we repeated the experiment in the presence of 1 mM MgCl_2_, where we expected a substantial portion of InsP_8_ to adopt the axial conformation (Figure 7b and Appendix A). We specifically chose assay conditions that allowed us to maintain constant pH, so it would not affect the electrostatic properties of the protein, while controlling the conformational equilibrium of InsP_8_ through the Mg^2+^ concentration alone. We observed that ΔH, ΔS and K_d_ all changed upon addition of MgCl_2_. K_d_ decreased to 205 nM (mean of three replicates). Interestingly, the enthalpy of binding became the dominant factor in the presence of MgCl_2_; ΔH became more negative (−5.9 kcal/mol) and −TΔS less negative (−3.3 kcal/mol) on average. These results confirm the negative correlation between ΔH and –TΔS, which is typical of protein ligand interactions. Enthalpically more favorable interactions are generally less favorable in terms of entropy, with a clear correlation across diverse ligand types [49]. K_d_ decreased notably, indicating improved binding in the presence of Mg^2+^ ions.

In summary, we found that addition of MgCl_2_ strengthened the binding of InsP_8_ to the VTC2 SPX-domain under approximately physiological conditions.

## 4. Discussion

We have conducted a systematic comparison of inositol poly- and pyrophosphates regarding their conformation at varying pH and ionic composition. Despite their structural similarities and comparable charge, clear differences between InsP_6_, 5PP-InsP_5_ and InsP_8_ became apparent. Generally speaking, the more phosphoryl groups the molecules carry, the lower the pH range in which the conformational change takes place. It was previously demonstrated for InsP_6_ that phosphoryl groups are spatially better separated in the axial conformation, minimizing steric and electrostatic repulsion between the anionic groups and providing one of the driving forces for the conformational change of higher InsPs (Figure 1c) [26,28]. Additional phosphoryl groups lead to accumulation of more negative charge in the equatorial plane, which means a critical charge density is reached at lower pH, facilitating the conformational change.

Our results also confirmed previous studies, which demonstrated that metal cation coordination can promote the transition to the axial conformation at lower pH. Frost et al. reported as early as 1979 that alkali cations stabilize the axial conformation of InsP_6_ [43]. There have since been multiple studies showing that complexation of various metals promotes the transition to axial InsP_6_ [47,50,51,52]. Hager et al. could subsequently show that Mg^2+^ ions help to stabilize 5PCP-InsP_5_ in an axial conformation [28,43]. Notably, we now found that InsP_8_ begins to change conformation at pH* 7.5 (=pH 7.4) in the presence of 5 equiv. MgCl_2_—reminiscent of cytosolic conditions—which led us to focus our subsequent efforts on this molecule.

Thermodynamic studies provided more information about the conformational equilibrium of InsP_8_. The transition from the equatorial to axial conformation is an exothermic process, and the formation of the axial conformer is enthalpically favored but entropically hindered. Addition of Mg^2+^ ions reduced both the enthalpic driving force and the entropic penalty, shifting the equilibrium more toward the axial side. As has been previously demonstrated for InsP_6_, the phosphoryl groups are more exposed to the solvent in the axial conformation, leading to a more ordered hydration shell and overall loss of entropy compared to equatorial conformation, hence a negative ΔS [26]. Positively charged Mg^2+^ ions will coordinate to the phosphoryl groups, thereby shielding some of the negative charge, which reduces hydration, and could explain why there is less entropic penalty associated with the conformational change at higher Mg^2+^ concentrations. Similarly, coordination of Mg^2+^ cations might also reduce electrostatic repulsion between phosphoryl groups in the equatorial plane, thus reducing the enthalpic driving force towards the axial conformation. Of course, the forces governing the conformational equilibrium are likely more complex than the interpretation above. A plethora of different multinuclear complexes can be expected to coexist and interconvert, each with its own protonation/complexation/conformation equilibria. The parameters we determined experimentally should therefore be understood as the sum of all these processes.

Trying to make a prediction about InsP_8_ in biological settings based on our results, we considered the following. Our assay conditions can only approximate cytosolic conditions, and we did not increase MgCl_2_ content beyond 0.25 mM or five equivalents relative to 50 µM InsP_8_, to avoid precipitation. Furthermore, the concentration of InsP_8_ could not be reduced below 50 µM, without compromising the NMR detection. However, the endogenous concentration InsP_8_ is thought to be around 1 µM, and cytosolic concentrations of free Mg^2+^ are approximately 0.5–1 mM [32,53]. Therefore, there is a far greater excess of Mg^2+^ in a cytosolic setting than in our assays. This excess should push the conformational equilibrium further toward the axial side than the ca. 30% we observed under our conditions. Nevertheless, five equivalents Mg^2+^ should largely saturate InsP_8_, which at pH 7.4 carries 10–11 negative charges (Figure 5). For comparison, InsP_6_ is also predicted to form pentamagnesium complexes under cytosolic conditions [24,46]. Taken together, these considerations strongly suggest that InsP_8_ is actively interconverting between conformations under cytosolic conditions. Based on our experiments, we expect more than 30% of the cytosolic pool of InsP_8_ to adopt an axial conformation under physiological conditions.

Phosphorous NMR studies revealed further details of the speciation of InsP_8_ regarding protonation and complexation with potassium and magnesium ions. At physiological pH and in the presence of K^+^ and an equimolar amount of Mg^2+^, the most abundant species is the equatorial complex [MgK_3_(H_3_L)]^6−^, in which Mg^2+^ is coordinated to P2, P1α and P1β. Another species present at physiological pH is the axial complex [MgK_3_(H_2_L)]^7−^ in which Mg^2+^ is coordinated by P5α and P1α. Overall, the pyrophosphate groups in positions 1 and 5 play an essential role in Mg^2+^ binding, as the ion is always coordinated to either the two pyrophosphate groups (in axial complexes) or P2 and PP1 (in equatorial complexes). Notably, all metal complexes of InsP_8_ are far more stable than the equivalent ones formed by InsP_6_ or 5PP-InsP_5_.

The Mg^2+^ complexes of PP-InsPs were also found to be more stable than those of ATP and ADP, two other well-known cytosolic Mg^2+^ chelators. Formation constants of complexes between Mg^2+^ and ATP, previously reported at 150 mM NaCl and 37 °C [54] are: Mg^2+^ + ATP^4−^ → [MgATP]^2−^: log(K) = 4.34, Mg^2+^ + HATP^3−^ → [Mg(HATP)]^−^: log(K) = 2.39. In contrast, we measured formation constants (log(K)) of 15.0 and 8.7 for [MgK_3_(H_3_L)] complexes of InsP_8_ and 5PCP-InsP_5_. It therefore appears feasible that the significant stability of these complexes influences their interactions with other biomolecules. Do metal ions bridge binding interactions between PP-InsPs and proteins? And if so, is this effect more pronounced for InsP_8_ than its less densely phosphorylated relatives? And how does this ultimately influence the strength and specificity of binding?

One could envision signaling functions associated with the metal complexation and the conformational equilibrium of PP-InsPs. In light of their role as cellular ATP and phosphate sensors, additional sensing functions seem plausible [8,48]. Given how sensitively PP-InsP speciation reacts to solution conditions in vitro, it is highly likely to change upon local subcellular perturbations in pH or metal composition, which may promote selective engagement of signaling partners. Such a pH-sensing mechanism has been demonstrated in the case of the yeast transcription factor Opi1, which is retained on the ER membrane by binding to phosphatidic acid (PA). Upon decrease of intracellular pH and protonation of the PA headgroup, Opi1 was released and activated its downstream target genes involved in inositol biosynthesis [55]. The authors proposed that phosphatidyl inositol lipids might sense pH through similar mechanisms, and the same might be true for soluble InsPs. Interestingly, the enzymes KCS1 and PLC1, part of the inositol pyrophosphate synthesis pathway in yeast, were found in a genome-wide screen for proteins involved in intracellular pH sensing [56].

In the large majority of cases, it is unclear by which mechanisms proteins manage to recognize and distinguish the different PP-InsPs [9,23]. Differential metal binding and/or a drastically altered molecular shape in the axial conformation might provide a convenient way to recognize the appropriate ligand. The idea that solution conditions, specifically the presence or absence of divalent cations, can have a significant influence on the outcome of in vitro binding studies with InsPs has already been proposed more than twenty years ago [57]. Our ITC experiments now provide a first hint that InsP speciation might play a role in this differentiation. Dissociation constants for InsP_8_ binding to the VTC2 SPX domain were almost cut in half by addition of 1 mM Mg^2+^, and binding shifted towards a more enthalpically dominated interaction in the presence of Mg^2+^. It is tempting to speculate that the decrease in ΔH might reflect a more specific fit of an axial magnesium complex into the binding site, compared to the free, equatorial molecule, but structural evidence would be a prerequisite to support this hypothesis and is currently unavailable. Based on the current data, we cannot exclude the possibility that Mg^2+^ might lower the K_d_ independently of InsP_8_ conformation, for example, by altering the enthalpy and entropy of desolvation upon formation of the InsP_8_-Mg complex. It will be interesting to investigate the influence of Mg coordination more systematically for other proteins targeted by PP-InsPs, such as the mammalian SPX domain of XPR1, or the C2 domains of synaptotagmin isoforms.

Overall, our results highlight the immense complexity of PP-InsP speciation and how this speciation may influence their behavior. Solution conditions for PP-InsP-protein binding experiments in the literature differ widely regarding pH, salts, and other additives. For future biochemical studies, it will be important to consider carefully which molecular species are formed under the given assay conditions, how those conditions might affect the experimental outcome, and to what extent the conditions mirror cellular settings. An accurate understanding of this complexity will be indispensable in decoding the diverse roles of InsPs and PP-InsPs as cellular messengers.

## Figures and Tables

**Figure 1 biomolecules-13-00645-f001:**
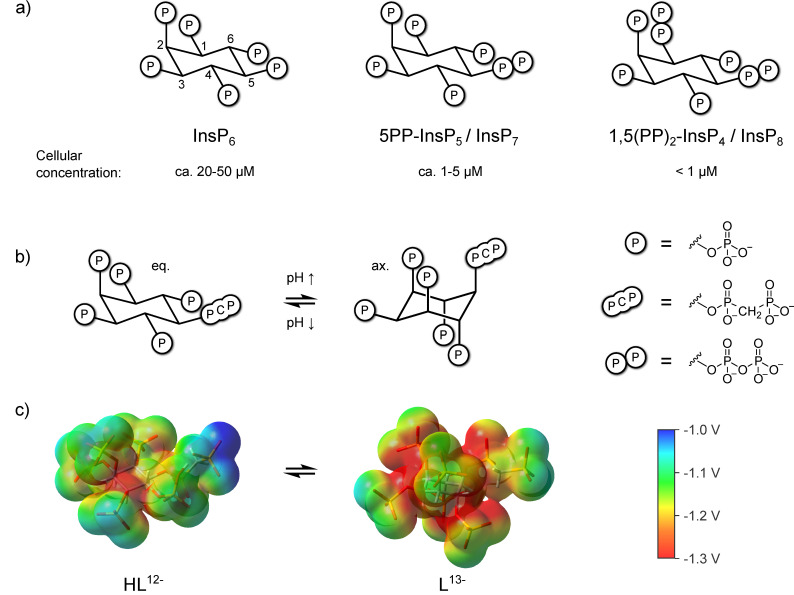
(**a**) Structure of InsP_6_, 5PP-InsP_5_, InsP_8_ and approximate concentrations in mammalian cells [29,30,31,32]. (**b**) pH-dependent equilibrium between axial (ax.) and equatorial (eq.) conformation of the non-hydrolyzable 5PP-InsP_5_ analogue 5PCP-InsP_5_ [28] (**c**) Electrostatic potential mapped on an isodensity surface for fully deprotonated, axial 5PCP-InsP_5_ (L^13−^) and monoprotonated, equatorial 5PCP-InsP_5_ (HL^12−^) (B3LYP/3-21 + G* geometries; isodensity value = 0.004, scale: −1.3 V (red) to −1.0 V (blue)). Atom color code: C (grey), H (white), O (red), P (orange).

**Figure 2 biomolecules-13-00645-f002:**
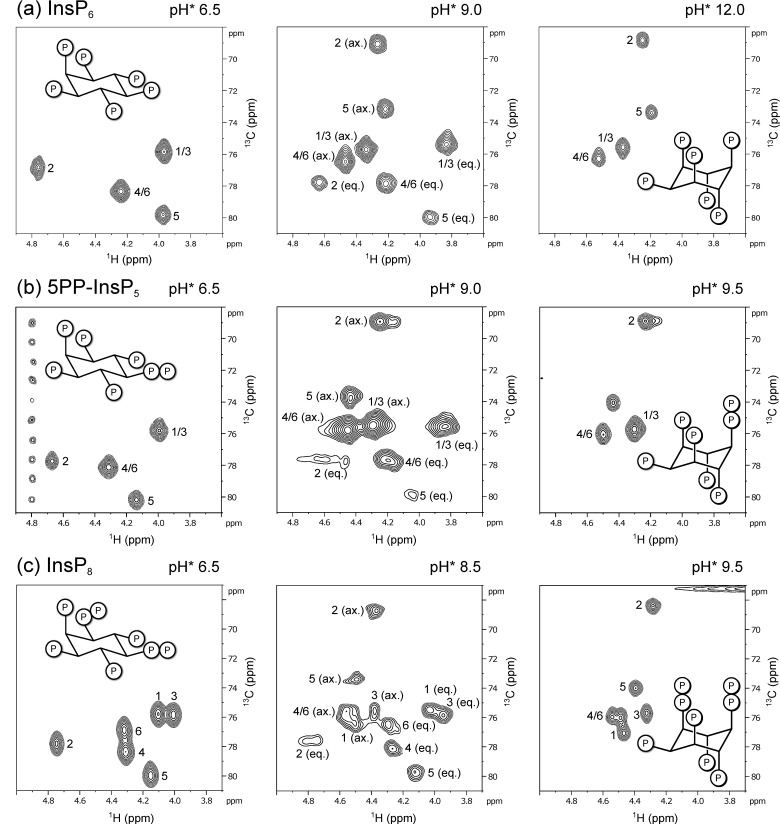
Representative ^1^H,^13^C-HMQC-NMR spectra of ^13^C-labeled (**a**) InsP_6_, (**b**) 5PP-InsP_5_ and (**c**) InsP_8_ in equatorial (eq.), axial (ax.) or mixed conformation, recorded at 4 °C. Peaks correspond to inositol backbone protons and are labeled according to their position in the myo-inositol ring. Samples contained 50 µM InsPs and 130 mM KCl, 10 mM NaCl in D_2_O. pH*-values measured in D_2_O can be converted to pH using the following formula: pH = 0.929 × pH* + 0.42 [35].

**Figure 3 biomolecules-13-00645-f003:**
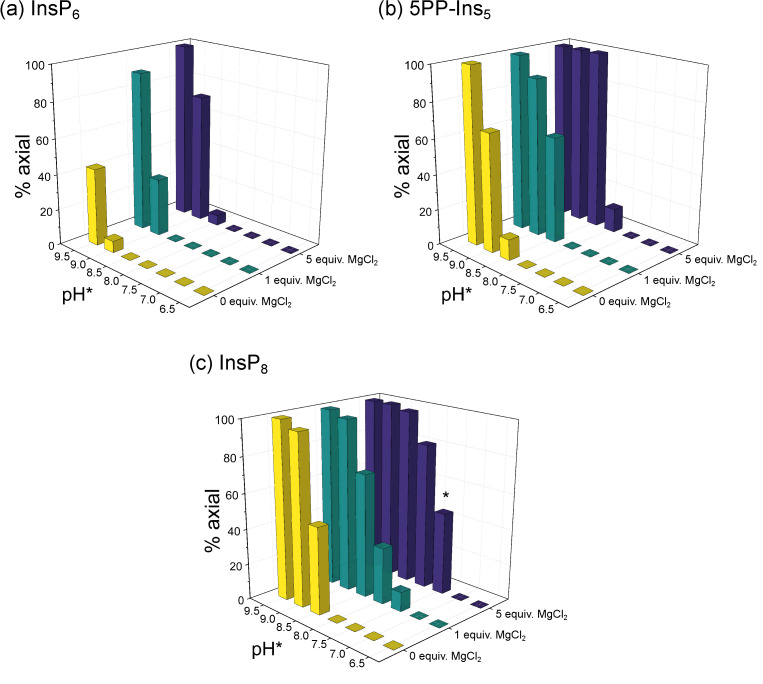
Relative abundance of (**a**) InsP_6_, (**b**) 5PP-InsP_5_ and (**c**) InsP_8_ in axial conformation at different pH and 50 µM total InsP concentration in the presence of 0/1/5 equivalents MgCl_2_ and physiological background of 130 mM KCl and 10 mM NaCl. ^1^H,^13^C-HMQC-NMR spectra were measured at 4 °C and integrated to obtain the ratio of eq. to ax. conformation. The proportion of ax. InsPs increases with pH, Mg^2+^ concentration and phosphorylation state (InsP_6_ < 5PP-InsP_5_ < InsP_8_). Intriguingly, about half of InsP_8_ molecules are in ax. conformation at physiological pH in the presence of five equivalents MgCl_2_ (bar marked with *). To enable NMR detection, all samples were made up in D_2_O. pH*-values measured in D_2_O can be converted to pH using the following formula: pH = 0.929 × pH* + 0.42 [35]. Peaks of both conformations were integrated to obtain the relative abundance of axial vs. equatorial conformer.

**Figure 4 biomolecules-13-00645-f004:**
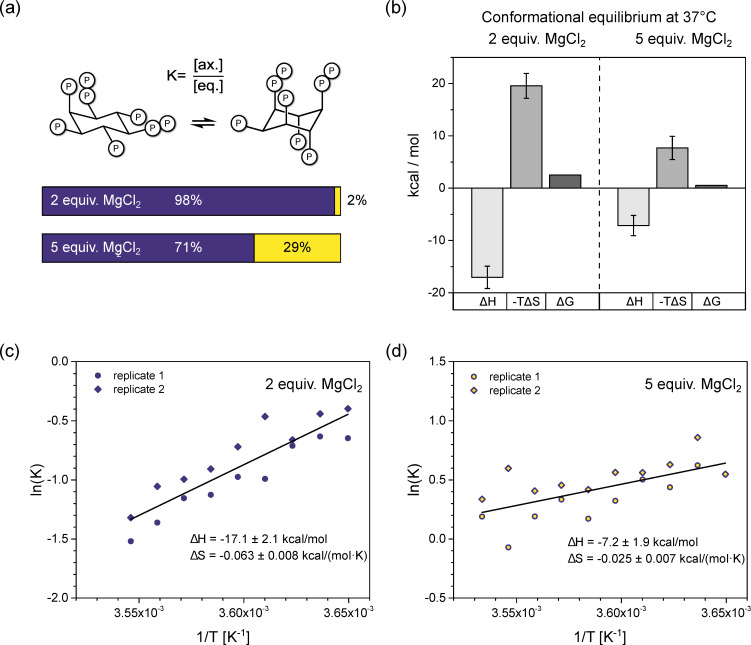
Thermodynamic analysis of InsP_8_ conformational equilibrium. (**a**) Proportions of axial vs. equatorial conformer in the presence of 2 equiv. vs. 5 equiv. Mg^2+^, according to Gibbs-Helmholtz equation and thermodynamic parameters from (**b**). (**b**) Results of van’t Hoff analysis. Transition from eq. to ax. conformation is an exothermic process. While both entropic and enthalpic contributions are decreased by additional Mg^2+^, overall, the equilibrium is shifted toward ax. conformation. (**c**,**d**) Van’t Hoff plots of InsP_8_ conformational equilibrium at pH 7.4 in the presence of (**c**) two equivalents, (**d**) five equivalents MgCl_2_ and physiological background of 130 mM KCl and 10 mM NaCl. Temperature range: 274–283 K. ^1^H,^13^C-HMQC-NMR spectra were integrated to obtain the ratio of eq. to ax. conformer, i.e., the equilibrium constant K of the conformational equilibrium. Two replicates were treated as one for the linear regression. Thermodynamic parameters are reported ± standard error of regression.

**Figure 5 biomolecules-13-00645-f005:**
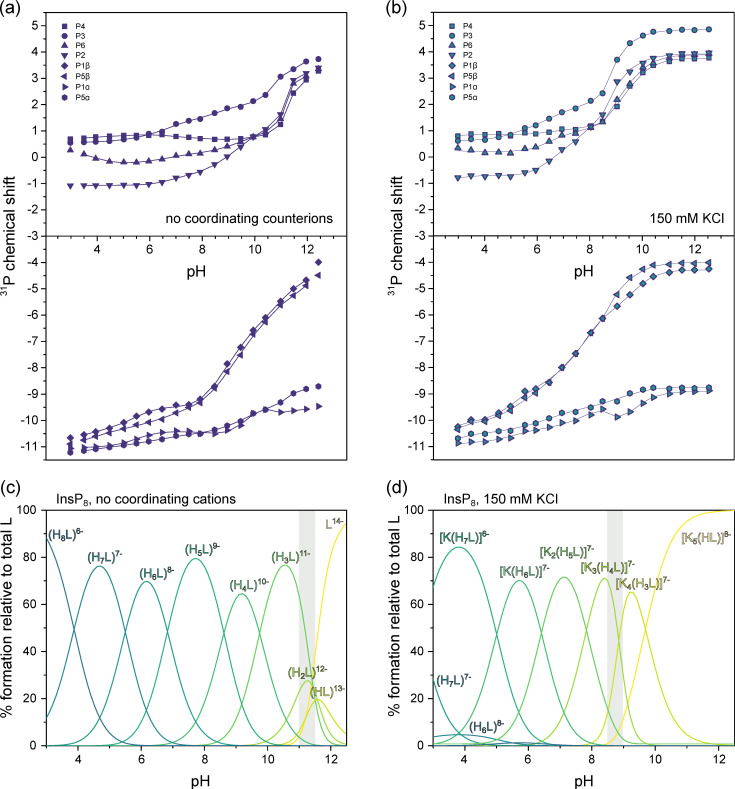
^31^P-NMR titrations of InsP_8_. (**a**) Chemical shift without coordinating counterions (150 mM NMe_4_Cl) at 22 °C, pH 3.0–12.5. (**b**) Chemical shift with 150 mM KCl. Dots represent experimental chemical shift data, separated into monophosphate (position 2, 3, 4, 6) and pyrophosphate groups (position 1 and 5). Lines represent theoretical chemical shift values based on the protonation model and calculated protonation constants. (**c**,**d**) Abundance of different protonation states (no coordinating counterions) or K^+^-complexes (in the presence of 150 mM KCl) of InsP_8_ (L) over the pH range. The grey area represents the pH-range with the biggest chemical shift changes in (**a**,**b**).

**Figure 6 biomolecules-13-00645-f006:**
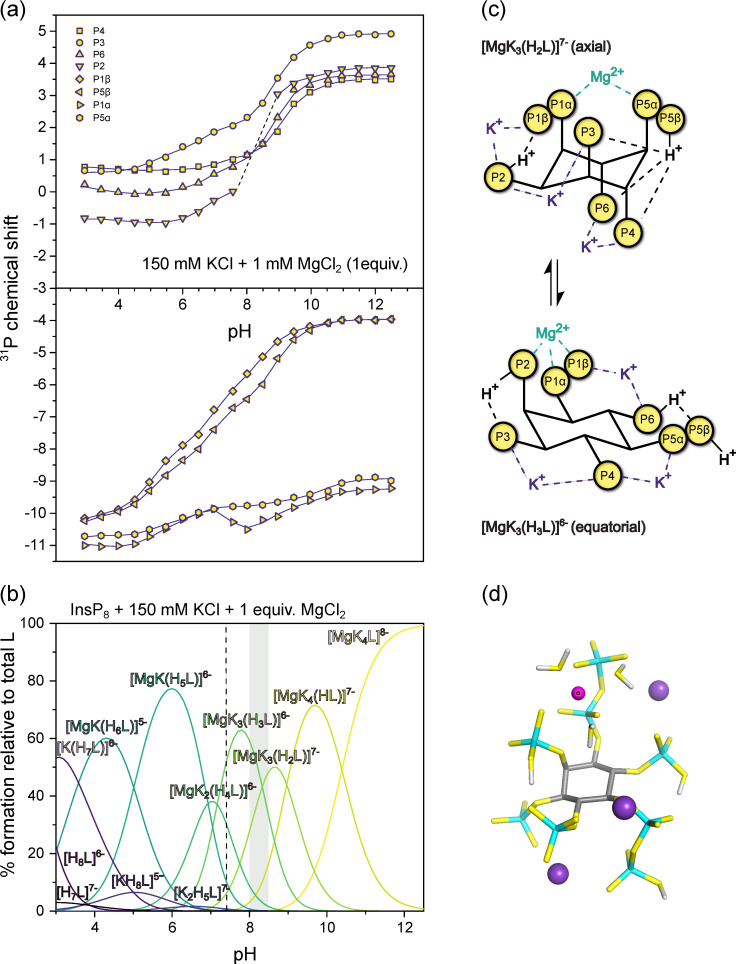
^31^P-NMR titrations of InsP_8_. (**a**) Chemical shift with 150 mM KCl and 1 mM MgCl_2_ at 22 °C, pH 3.0–12.5. Dots represent experimental chemical shift data, separated into monophosphate (position 2, 3, 4, 6) and pyrophosphate groups (position 1 and 5). Lines represent theoretical chemical shift values based on the protonation model and calculated protonation constants. (**b**) Abundance of different or K^+^-Mg^2+^-complexes (in the presence of 150 mM KCl) of InsP_8_ (L) over the pH range. pH 7.4 is indicated by a dashed line. The grey area represents the pH-range with the biggest chemical shift changes in a. (**c**) Schematic view of the highest protonated axial and the lowest protonated equatorial complexes [MgK_3_(H_2_L)]^7−^ and [MgK_3_(H_3_L)]^6−^ (**d**) DFT optimized structure of the most abundant complex at pH 7.4, [MgK_3_(H_3_L)]^6−^.

**Figure 7 biomolecules-13-00645-f007:**
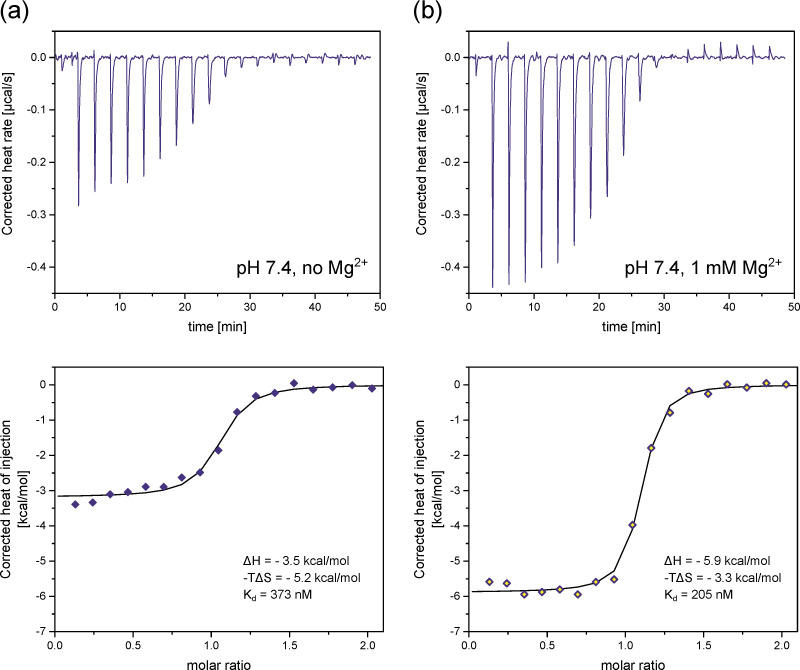
Isothermal titration calorimetry of (**a**) InsP_8_ binding to the yeast VTC2 SPX–domain (50 µM) in ITC binding buffer (25 mM HEPES pH 7.4, 150 mM KCl, 40 mM NaCl, 0.5 mM TCEP) at 25 °C. (**b**) Addition of 1 mM MgCl_2_ substantially decreases ΔH, ΔS and the binding constant K_d_. The figure shows one representative replicate per condition, out of three replicates (Appendix A). ΔH, ΔS and K_d_ are reported as the average of three replicates.

**Table 1 biomolecules-13-00645-t001:** Logarithms of the protonation and formation constants of InsP_8_ (*I* = 0.15 M; *T* = 22 °C).

Equilibrium	log *K*		
	InsP_8_	5PCP-InsP_5_	InsP_6_
	^31^P NMR (22 °C) ^a^	^31^P NMR (22 °C) ^b^	Potentiometry (37 °C) ^b^
L^14−^ + H^+^ ↔ HL^13−^	11.21(1)	11.48(1)	10.8(1)
L^14−^ + 2 H^+^ ↔ H_2_L^12−^	22.78(2)	22.42(2)	21.3(1)
L^14−^ + 3 H^+^ ↔ H_3_L^11−^	34.22(2)	32.26(2)	31.63(6)
L^14−^ + 4 H^+^ ↔ H_4_L^10−^	43.96(2)	40.94(2)	40.42(6)
L^14−^ + 5 H^+^ ↔ H_5_L^9−^	52.58(3)	47.67(2)	47.32(6)
L^14−^ + 6 H^+^ ↔ H_6_L^8−^	59.41(8)	51.93(2)	53.04(7)
L^14−^ + 7 H^+^ ↔ H_7_L^7−^	64.91(6)	55.64(2)	56.14(9)
L^14−^ + 8 H^+^ ↔ H_8_L^6−^	68.79(7)	---	---
5 K^+^ + HL^13−^ ↔ [K_5_(HL)]^8−^	12.47(3)	6.57(3)	---
4 K^+^ + H_2_L^12−^ ↔ [K_4_(H_2_L)]^8−^	9.76(3)	4.61(3)	---
4 K^+^ + H_3_L^11−^ ↔ [K_4_(H_3_L)]^7−^	---	4.50(5)	5.42(5)
3 K^+^ + H_4_L^10−^ ↔ [K_3_(H_4_L)]^7−^	5.446(5)	3.94(4)	3.36(5)
2 K^+^ + H_5_L^9−^ ↔ [K_2_(H_5_L)]^7−^	3.820(7)	2.79(7)	---
K^+^ + H_6_L^8−^ ↔ [K(H_6_L)]^7−^	2.58(5)	---	---
K^+^ + H_7_L^7−^ ↔ [K(H_7_L)]^6−^	2.08(3)	---	---
Mg^2+^ + L^14−^ + 4 K^+^ ↔ [MgK_4_L]^8−^	22.4(1)	---	---
Mg^2+^ + HL^13−^ + 4 K^+^ ↔ [MgK_4_(HL)]^7−^	21.6(1)	11.64(5)	---
Mg^2+^ + H_2_L^12−^ + 3 K^+^ ↔ [MgK_3_(H_2_L)]^7−^	18.1(1)	9.75(7)	---
Mg^2+^ + H_3_L^11−^ + 3 K^+^ ↔ [MgK_3_(H_3_L)]^6−^	15.0(1)	8.79(7)	---
Mg^2+^ + H_4_L^10−^ + 2 K^+^ ↔ [MgK_2_(H_4_L)]^6−^	11.6(1)	6.96(7)	---
Mg^2+^ + H_5_L^9−^ + K^+^ ↔ [MgK(H_5_L)]^6−^	9.1(1)	5.26(7)	---
Mg^2+^ + H_6_L^8−^ + K^+^ ↔ [MgK(H_6_L)]^5−^	7.3(1)	---	---
Mg^2+^ + H_6_L^8−^ ↔ [Mg(H_6_L)]^6−^	---	4.71(7)	---

^a^ This work, σ = 0.033 (H^+^), 0.053 (K^+^) and 0.051 (Mg^2+^). ^b^ Thermodynamic data reported previously for similar systems are included for comparison; the equations in the left column refer to InsP_8_ complexes [28,46,47]. The standard deviation on the uncertain digit is added between brackets.

## Data Availability

The raw data presented in this study are openly available on the Zenodo repository under the doi:10.5281/zenodo.7665634.

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
