# Peer review of "One Scaffold, Two Conformations: The Ring-Flip of the Messenger InsP_8_ Occurs under Cytosolic Conditions"

_biomolecules, 2023, doi:10.3390/biom13040645_

Round 1
Reviewer 1 Report
The authors use 13C-NMR to study the conformational states of inositol phosphates (InsPs), which are a ubiquitous cellular signalling molecule. They use NMR to confirm that InsPs adopt both axial and equatorial conformations in solution. They further indicate that these exchanges happen in cytoplasmic conditions and that this is influenced by factors such as pH, cations, and salt. The authors also show Mg2+ can influence the affinity of InsP8 for target proteins, suggesting that InsP8 can act as a molecular switch. This manuscript provides important and fundamental characterization of how InsPs signalling can be modulated by the presence of other ions and gives important context for other studies of these biomolecules.
This manuscript is well-written and interesting. Experiments are carried out and analyzed properly. This manuscript is suitable for publication in Biomolecules and I only have a minor question:
Minor points:
The authors detail in the introduction that “phosphate efflux via XPR1 is activated almost exclusively by InsP8, even though InsP6 and 5PP-InsP5 bind with relatively similar dissociation constants (Kd) to the SPX domain”
The authors have measured the affinity of InsP8 to SPX via ITC and notice a change in affinity due to Mg2+. Does a similar change occur with InsP8 and could this explain the activation of XPR1?
Author Response
The authors detail in the introduction that “phosphate efflux via XPR1 is activated almost exclusively by InsP8, even though InsP6 and 5PP-InsP5 bind with relatively similar dissociation constants (Kd) to the SPX domain”
The authors have measured the affinity of InsP8 to SPX via ITC and notice a change in affinity due to Mg2+. Does a similar change occur with InsP8 and could this explain the activation of XPR1?
Response:
The reviewer raises a very interesting point. To the best of our knowledge, the SPX domain from XPR1 is the only other SPX domain, for which an InsP8 binding constant has been reported in the literature (Shears and coworkers, PNAS 2020). The published ITC binding curves were recorded in the presence of 800 mM MgCl2, at pH 7.2, and the Kd of InsP8 was determined to be 180 nM, while the Kd of 5PP‑InsP5 amounted to 340 nM. It would be very interesting to find out in future studies, if slightly higher pH (favoring the axial conformation of InsP8) can lead to a more pronounced difference in Kd, and how binding constants change in the absence of Mg2+. We have included a sentence in the discussion where we point out the need to investigate PP-InsP protein interactions more systematically, and in the presence and absence of Mg2+.
Reviewer 2 Report
This manuscript by Kurz et alteri describes an NMR- and ITC-based study of conformational changes of three inositol poly/pyrophosphates under changing conditions such as pH. These changes are brought about by changes in protonation levels, and complexation with K+ and Mg2+, as investigated by NMR titration experiments. Finally, using ITC, it is shown that under high Mg2+ conditions InsP8 has a higher affinity for its binding partner SPX domain, presumable because the Mg2+ pushed the conformational equilibrium more towards the binding-prone axial conformation, away from the equatorial conformation.
I enjoyed reading the paper as it is very clearly written. The findings are significant because the uncovered mechanism may be the basis for a molecular switch, which may be further investigated in the future.
There is an uncertainty as to the exact interpretation of the order and location of protonation and complexation events on InsPs. However, the authors clearly point out these uncertainties and go into depth to rationalize every conclusion they draw, including providing alternative, but less likely, interpretations, in an extended appendix.
Therefore, I recommend publication of this work in Biomolecules.
The following points have to be addressed:
- How do you know that the change in the structure of InsP and not simply the presence of Mg2+ changes the affinity to VTC2 SPX domain? Maybe Mg2+ occupies a location on the SPX domain in the interaction interface. The altered affinity may also be a combination of both mechanisms.
- I wonder if direct 13C-detection would circumvent peak broadening observed for 1H. This is usually the case for proteins. Your molecules are 13C-labeled and according to the 2D experiments the resonances are sufficiently dispersed to resolved them in 1D spectra.
- I do not understand the assay in Fig. A3. If the intrinsic peak intensity is lower at 6 pH (understandable), how can you judge if you gained everything back at 6 pH? What is measured for the control?
- Also Fig. A3: There seem to be two identical boxes for pH 6 after filtration.
- Page 8, line 257: This should refer to figure A3.
- Page 7, line 244: S1 should be A1.
- Fig. A2 is not referenced in the text (probably should be done in Section 3.1).
- I found it cumbersome to have only numeric values for pH* and given a formula. It’d be helpful to have the pH directly reported where appropriate.
- For Fig. 3: I think I read it at some point but I can’t find it anymore: How many peaks did you use to obtain the peak intensities (or volumes)? If more than one, which is recommended, you can calculate error bars.
- Page 8, line 274: Did you really mean to write “by increasing the degree of phosphorylation”?
- Page 8, line 279: Fig. A22 should be Fig. A2 (and also later analogously). Why is this only mentioned in the context of InsP8, but the figure also contains data for Ins6 and 5PP-InsP5?
- Why only 2 ITC replicates and not 3?
Author Response
- How do you know that the change in the structure of InsP and not simply the presence of Mg2+ changes the affinity to VTC2 SPX domain? Maybe Mg2+ occupies a location on the SPX domain in the interaction interface. The altered affinity may also be a combination of both mechanisms.
This is an important point raised by the reviewer. With our current data, we cannot say with certainty that the ring-flip is responsible for the change in affinity. While the ring-flip is one possible explanation, another explanation is the metal coordination between the InsP and the protein (as pointed out by the reviewer). Yet another interpretation could be that the hydration shell of the InsP8-Mg complex is altered, which will change the enthalpy and entropy of desolvation upon binding.
We would like to point out that our lab has tried - for quite some time – to observe protein-bound inositol pyrophosphates using 13C-labeled ligands and NMR spectroscopy. These experiments should, in principle, be able to assign the bound ligand conformation. Unfortunately, in all cases investigated to date, the ligand exchange rate is in the intermediate exchange regime, leading to severe line broadening and making the bound state invisible.
- I wonder if direct 13C-detection would circumvent peak broadening observed for 1H. This is usually the case for proteins. Your molecules are 13C-labeled and according to the 2D experiments the resonances are sufficiently dispersed to resolved them in 1D spectra.
Thank you for bringing this up. There are two reasons why we decided against direct 13C detection. Firstly, we wanted to keep concentrations as low as possible to approximate cellular concentrations, and the sensitivity of the probe head towards 13C is lower than towards 1H.
Secondly, and more importantly, our material is 13C labeled in all six positions, which would lead to extensive C-C coupling and overlapping multiplets in the C-dimension, especially when both conformations are present. In order to allow quantification, we therefore opted for the better resolution of 2D spectra, and accepted the need for lower temperatures.
- I do not understand the assay in Fig. A3. If the intrinsic peak intensity is lower at 6 pH (understandable), how can you judge if you gained everything back at 6 pH? What is measured for the control?
The control experiments are samples of the same concentration but set to pH 6 from the beginning. We quantify intensity relative to the same internal standard (myo-inositol) in both the controls and the pH9 samples, then filter, and subsequently adjust all samples to pH6 and measure again. The assumption is that no precipitation occurs in the controls because of the low pH; the other samples should show the same intensity at the end, unless there was precipitation.
We have changed the coloring and the labels in the graph to try and avoid any confusion.
- Also Fig. A3: There seem to be two identical boxes for pH 6 after filtration.
Thank you for catching this. The legend was indeed doubled, and we have fixed it now.
- Page 8, line 257: This should refer to figure A3.
Yes, thank you for finding this error, we have fixed the label.
- Page 7, line 244: S1 should be A1.
Yes, again, thank you for catching this mistake, the label has been changed.
- Fig. A2 is not referenced in the text (probably should be done in Section 3.1).
Thank you for pointing this out. We have now included a reference for Figure A2.
- I found it cumbersome to have only numeric values for pH* and given a formula. It’d be helpful to have the pH directly reported where appropriate.
We agree, that the unit pH* is not necessarily the most user-friendly. Nevertheless, we feel that it is important to report pH*, since we used a pH meter in deuterated solvents. To improve the readability, we have included the calculated pH values in several parts of the text.
- For Fig. 3: I think I read it at some point but I can’t find it anymore: How many peaks did you use to obtain the peak intensities (or volumes)? If more than one, which is recommended, you can calculate error bars.
We typically only use one peak for integration, because in several cases integration is not straightforward due to overlapping resonances, or close proximity to the water peak.
- Page 8, line 274: Did you really mean to write “by increasing the degree of phosphorylation”?
We have rephrased this sentence, to avoid any confusion, thank you.
- Page 8, line 279: Fig. A22 should be Fig. A2 (and also later analogously). Why is this only mentioned in the context of InsP8, but the figure also contains data for Ins6 and 5PP-InsP5?
Thank you for catching the mislabeled legend, this has been fixed. We have also changed where we reference this data in the main text, so that it is clear these experiments were done for InsP6 and 5PP-InsP5 as well.
- Why only 2 ITC replicates and not 3?
We have repeated the ITC runs and are now reporting triplicates, thank you. Figure 7 now shows one representative replicate, all three replicates can be found in the newly added figure A9.